# Synthesis of Bis(amino acids) Containing the Styryl-cyclobutane Core by Photosensitized [2+2]-Cross-cycloaddition of Allylidene-5(4*H*)-oxazolones

**DOI:** 10.3390/ijms24087583

**Published:** 2023-04-20

**Authors:** Sonia Sierra, David Dalmau, Juan V. Alegre-Requena, Alexandra Pop, Cristian Silvestru, Maria Luisa Marín, Francisco Boscá, Esteban P. Urriolabeitia

**Affiliations:** 1Instituto de Síntesis Química y Catálisis Homogénea (ISQCH), CSIC—Universidad de Zaragoza, Pedro Cerbuna 12, 50009 Zaragoza, Spain; ssierrasainzaja@gmail.com (S.S.); ddalmau@unizar.es (D.D.); 2Supramolecular Organic and Organometallic Chemistry Centre (SOOMCC), Department of Chemistry, Faculty of Chemistry and Chemical Engineering, Babeş-Bolyai University, 400028 Cluj-Napoca, Romania; alexandra.m.pop@ubbcluj.ro (A.P.); cristian.silvestru@ubbcluj.ro (C.S.); 3Instituto Universitario Mixto de Tecnología Química (ITQ-UPV), Universitat Politècnica de València-Consejo Superior de Investigaciones Científicas, 46022 València, Spain; marmarin@qim.upv.es (M.L.M.); fbosca@itq.upv.es (F.B.)

**Keywords:** amino acids, truxinic, photocatalysis, cyclobutane, sensitization, laser flash photolysis, density functional theory (DFT) calculations, triplet excited state

## Abstract

The irradiation of 2-aryl-4-(*E*-3′-aryl-allylidene)-5(4*H*)-oxazolones **1** with blue light (456 nm) in the presence of [Ru(bpy)_3_](BF_4_)_2_ (bpy = 2,2′-bipyridine, 5% mol) gives the unstable cyclobutane-bis(oxazolones) **2** by [2+2]-photocycloaddition of two oxazolones **1**. Each oxazolone contributes to the formation of **2** with a different C=C bond, one of them reacting through the exocyclic C=C bond, while the other does so through the styryl group. Treatment of unstable cyclobutanes **2** with NaOMe/MeOH produces the oxazolone ring opening reaction, affording stable styryl-cyclobutane bis(amino acids) **3**. The reaction starts with formation of the T_1_ excited state of the photosensitizer ^3^[Ru*(bpy)_3_]^2+^, which reacts with S_0_ of oxazolones **1** through energy transfer to give the oxazolone T_1_ state ^3^(oxa*)-**1**, which is the reactive species and was characterized by transient absorption spectroscopy. Measurement of the half-life of ^3^(oxa*)-**1** for **1a**, **1b** and **1d** shows large values for **1a** and **1b** (10–12 μs), while that of **1d** is shorter (726 ns). Density functional theory (DFT) modeling displays strong structural differences in the T_1_ states of the three oxazolones. Moreover, study of the spin density of T_1_ state ^3^(oxa*)-**1** provides clues to understanding the different reactivity of 4-allylidene-oxazolones described here with respect to the previously reported 4-arylidene-oxazolones.

## 1. Introduction

The role of amino acids as key building blocks in the synthesis of peptides and proteins is critical [1,2]. Each parameter of their three-dimensional structure, as well as their electronic and steric characteristics, must be carefully designed to produce the necessary effect within the structure of the protein that hosts them. Given their relevance, it is clear that having versatile and selective synthetic methods for the preparation of amino acid analogues of natural ones is highly desirable. In fact, interest in new methods for the synthesis of unnatural amino acids derives from the possibility of introducing small controlled changes that lead to unusual structural situations and that allow modulation of the activity of the proteins [3,4,5,6,7,8,9,10].

The synthesis of truxillic and truxinic acids, originally isolated from plant extracts [11,12], has experienced growing interest in recent years due to their pharmacological properties (Figure 1a). In this respect, they show strong anti-inflammatory and antinociceptive activities through inhibition of different targets, such as fatty acid binding protein (FABP), anoctamin-1, NO, and radicals [13,14,15,16,17,18,19,20]. They also show antioxidant and anti-hypoglycaemic activities [21] and cytotoxic activity [22]. In addition, they are involved in glucose metabolism through activation of peroxisome proliferator-activated receptors (PPAR) [23,24]. Beyond biological activities, they also show interesting properties as internal donors in Ziegler–Natta catalysts [25].

Closely related to the truxillic and truxinic acids are the unnatural bis-amino acids 1,3-diaminotruxillic and 1,2-diaminotruxinic derivatives (Figure 1b), which also show antinociceptive activity, and, moreover, are promising substrates in the treatment of diabetes type-II because they are the only non-peptidic agonists of the Glucagon Like-Peptide-1 (GLP-1) receptor [26,27,28]. The basic scaffold of 1,3-diaminotruxillic and 1,2-diaminotruxinic amino acids is a cyclobutane; therefore, the simplest synthetic approximation for their preparation could be the [2+2]-photocycloaddition of two olefins [29], more specifically, two 4-aryliden-5(4*H*)-oxazolones (Figure 1b). Despite its apparent simplicity, this synthetic method has been scarcely investigated, and, as far as we know, only four reports have appeared in the literature (Figure 2).

Between 2007 and 2012, Wang and coworkers reported two examples of [2+2]-photocycloaddition of oxazolones, involving irradiating oxazolone for 3 days with a high-power Hg lamp (500 W); they obtained the corresponding 1,3-diaminotruxillic cyclobutanes as four different isomers with a global yield of 10% [26,27,28]. In 2017, our group improved both the scope and the yield of the reaction using low-power (20 W) LED blue light (465 nm), but still a mixture of diaminotruxillic derivatives was obtained [30] (Figure 2a). Further contributions from Amarante [31] and our group [32] have included the synthesis of 1,2-diaminotruxinic derivatives by irradiation of aryliden-oxazolones with blue light in the presence of a photocatalyst ([Ru(bpy)_3_](BF_4_)_2_, versus Eosin Y, Figure 2b,c). Remarkably, the reaction in the presence of the photocatalyst takes place with a different orientation with respect to that in its absence (1,2- instead of 1,3-coupling) and with complete stereoselectivity, since the corresponding 1,2-diaminotruxinic derivatives were obtained as single isomers. This fact is quite noteworthy because this methodology allows control of the regio- and diastereoselectivity, so the method is simple, cheap, and efficient. The geometric isomers obtained were different as a function of the photocatalyst since the μ-isomer was obtained in the case of the Ru-photosensitized reaction, while the *zeta*-isomer was obtained for the eosin Y photocatalyst. All these observations give an idea of the potential of this known methodology to be applied to the synthesis of bis-amino acids using oxazolones as starting materials.

However, as mentioned previously, these processes have only been studied using 4-aryliden-5(4*H*)-oxazolones as precursors. These substrates contain only one C=C bond able to be activated, so the number of isomers of the bis-amino acid is limited. The introduction of additional activatable C=C bonds in the molecule would give access to new bis-amino acids not achievable by other synthetic methods. In addition, the possibility to control regio- and stereoselectivity, as the number of C=C double bonds increases, is highly challenging.

Here, we report the results obtained from the irradiation of 2-aryl-4-(*E*-3′-aryl-allylidene)-5(4*H*)-oxazolones **1**, in solution and in the presence of the [Ru(bpy)_3_](BF_4_)_2_ photocatalyst. These oxazolones contain an additional C=C bond conjugated with the exocyclic C=C bond, resulting in different reactivity. As well as the structural characterization of the resulting bis-amino acids, the determination of the photochemical active species was accomplished by transient absorption spectroscopy and density functional theory (DFT) modeling. The theoretical studies also provide clues for the understanding of the different reactivity found here.

## 2. Results and Discussion

### 2.1. Synthesis of the 2-Aryl-4-(E-3′-aryl-allylidene)-5(4H)-oxazolones ***1a***–***1h***

The oxazolones **1a-1h**, shown in Figure 3, were prepared following the classical Erlenmeyer–Plöch method [33,34,35,36,37,38,39,40]. The cinnamaldehydes and the hippuric acids employed in the synthesis were selected in order to cover a wide set of possibilities, with electron-releasing and electron-withdrawing substituents in the two aryl rings and in the allyl group. Oxazolones **1a**, **1c**, and **1d** appear on SciFinder, but **1c** and **1d** have not been characterized. Therefore, they are fully characterized here. For oxazolone **1a,** data were given [41] but in different solvents (CDCl_3_, dmso-d_6_) than that reported here (toluene-d_8_). Oxazolones **1b**, **1e-1h** are new compounds and were fully characterized by NMR, MS, and X-ray diffraction data. All the oxazolones were obtained in moderate to good yields, except oxazolone **1g** which is highly soluble in ethanol (see Materials and Methods for details).

NMR data (Appendix A) showed that oxazolones **1d** and **1e** (Appendix A) were obtained as single isomers. The configuration of the C=C double bonds of the C(H)=C(Me)-C(H)=C fragment can be clearly inferred in the ^1^H-selective NOESY of **1d**. The perturbation of the signal at 7.16 ppm (H_3’_, see Figure 3 and Appendix A) produced a strong NOE in the signal at 7.07 ppm (H_1’_), as well as in the signal assigned to the *ortho* H of the 3′-Ph ring (see also Materials and Methods), showing their proximity. This means that the configuration of the C3’=C2’ double bond is *E* in **1d**. The same conclusions can be derived from the observation in **1e** of a NOE between the peaks at 8.44 ppm (H_3’_) and 7.08 ppm (H_1’_), although in this case, due to the presence of the Br atom, the configuration is *Z*. The configuration of the exocyclic C1’=C4 double bond was determined, in turn, by measurement of the ^3^J_CH_ coupling constant of the carbonyl signal with H_1’_ in the ^13^C NMR spectrum of **1d**. The value determined (5.2 Hz) shows that this C=C bond has a *Z*-configuration by comparison with reported data [42,43]. Therefore, **1d** and **1e** show *EZ* and ZZ configurations, respectively (Figure 3).

However, analysis of the NMR data in toluene-d_8_ of **1a**–**1c** (Appendix A) and **1f**–**1g** (Appendix A) showed the presence of a mixture of two isomers in 3.3/1 to 1.7/1 molar ratios, depending on the oxazolone. Two sets of signals assigned to the protons of the -C3’(H)=C2’(H)-C1’(H)=C4 fragment were observed, with ^3^J_H3’H2’_ values identical in the two isomers (15.7 Hz), suggesting an *E* configuration of the C3’=C2’ double bond in both isomers. As expected, the large ^3^J_H1’H2’_ value (11.6 Hz) suggests the formation of the s-trans rotamer of the diene system. Thus, the source of isomerism in these oxazolones must be located in the relative disposition of the oxazolone ring with respect to the C1’=C4 bond. The proton-coupled ^13^C NMR spectrum of **1a** showed the presence of two doublet peaks assigned to the CO group. The major species showed a value of the coupling constant ^3^J_CH_ = 4.2 Hz. According to previous reports [42,43], this suggests that the C1’=C4 bond has a *Z*-configuration; therefore, the major isomer of the oxazolone is *EZ*. As expected, in the minor isomer, this value was ^3^J_CH_ = 11.6 Hz, suggesting an *E*-configuration for this bond and that the oxazolone is *EE*.

The determination of the crystal structure of **1a** provides additional information. A molecular drawing is shown in Figure 4, while relevant crystallographic data and selected bond distances and angles are given as Appendix A.

The structure shows that the two C=C double bonds (C2-C10 and C11-C12) are *trans*, thus corresponding to the minor *EE*-isomer. The allylidene fragment C13-C12-C11-C10-C2 shows the expected long-short-long-short pattern of bond distances, corresponding to a C-C=C-C=C- molecular skeleton. Despite this pattern, the individual values of the bond distances show electronic delocalization throughout this system, as both the long bond distances (C12-C13 = 1.455(4) Å and C10-C11 = 1.427(4) Å), as well as the short ones (C11-C12 = 1.358(4) Å and C2-C10 = 1.361(4) Å) are in the respective range of distances found in the literature for this type of conjugated bonds [44]. Other internal parameters of the oxazolone and Ph rings do not show deviations with respect to similar compounds found in the literature [45,46,47,48].

### 2.2. Synthesis of the Cyclobutane-bis(oxazolones) ***2*** and 1,2-Diaminotruxinic Bis-amino Acids ***3*** by [2+2]-Photocycloaddition of Oxazolones 1

The irradiation of oxazolones **1a**–**1h** in CD_2_Cl_2_ at room temperature with the blue light provided by a Kessil lamp (456 nm) was monitored by ^1^H NMR and showed the appearance of many peaks in the 4–6 ppm region. This fact suggests the formation of the expected cyclobutanes, but the large number of peaks observed suggests the formation of many isomers and that the reaction takes place without selectivity. We previously observed that the introduction of a photosensitizer improved the selectivity of the reaction [32]. In this respect, the irradiation of oxazolones **1a, 1b, 1d**, and **1f** with blue light (456 nm, Kessil lamp) in deoxygenated CD_2_Cl_2_ at room temperature, under Ar atmosphere and in the presence of [Ru(bpy)_3_](BF_4_)_2_ (5% mol), took place with complete conversion of the oxazolone **1** in 18 h and formation of the corresponding cyclobutanes **2a**, **2b**, **2d** and **2f** (Figure 5). The characterization of these cyclobutanes by NMR spectroscopy (Appendix A) showed that each one was formed by a mixture of two isomers, one of them clearly as a major species with respect to the other (molar ratio 10.1/1 for **2a**, 3.6/1 for **2b**, 2.9/1 for **2d**, 4.8/1 for **2f**). For **2a** and **2f** this mixture could be separated by flash column chromatography. The structure determined from NMR data for isolated **2a** and **2f** is shown in Figure 5. For **2b** and **2d**, the mixture could not be separated, and the structure represented in Figure 5 was assigned to the major isomer. The structure of the minor isomers could not be determined due to their low molar ratio.

The formation of only two isomers for cyclobutanes **2a**, **2b**, **2d**, and **2f**, one of them in a clear majority, is remarkable (i.e., more than 80 possible isomers) and shows that the photosensitized reactivity of allyliden-5(4*H*)-oxazolones **1** also has a high degree of selectivity. Despite this, the scope of this reaction was more limited, since not all attempted oxazolones showed clear reactivity. The monitoring of the irradiation of **1c**, **1e**, **1g**, and **1h** under the same reaction conditions (456 nm, Ru 5%, CH_2_Cl_2_, Ar) by ^1^H NMR showed complete conversion of the oxazolones **1**, but also the presence of many peaks in the 4–6 ppm region, suggesting the formation of many different isomers of the respective cyclobutanes **2c**, **2e**, **2g** and **2h**. These mixtures proved to be intractable and were not further analyzed.

The cyclobutanes **2a**, **2b**, **2d**, and **2f** were stable once isolated as solids, but, in solution at room temperature, they underwent a thermal retro [2+2] reaction giving back the starting oxazolones **1a**, **1b**, **1d**, and **1f**. This low stability is related to intramolecular steric strains. It was also observed in μ-truxinic derivatives [32], and it seems to be responsible for the low yields observed after chromatographic purification (Figure 5). The simplest strategy to eliminate this steric constraint and to obtain stable cyclobutane derivatives is the transformation of the oxazolone ring in compound **2** into the corresponding ester **3** (Figure 6). The reaction can be performed in a one-pot, two-steps way, without isolating the cyclobutane intermediate **2**, therefore minimizing the erosion of yield during isolation of **2**.

This strategy was exemplified in the cases of **1a** and **1b**, as represented in Figure 6. The irradiation of **1a** and **1b** in deoxygenated CH_2_Cl_2_ in the presence of the Ru-photosensitizer was performed as reported above. Once the full conversion of **1a,b** was observed (18 h), the reaction solvent was evaporated to dryness, while irradiation was kept to minimize the retro [2+2] reaction in intermediates **2**. The dry residues were suspended in methanol and subjected to ring opening reaction by treatment with NaOMe and heating, giving the 1,2-diaminotruxinic acids **3a** and **3b** in good to excellent yields (Figure 6). As is evident from Figure 6, the improvement in the reaction yields was more than notable. As expected, both **3a** and **3b** were stable in solid and in solution and could be purified, crystallized, and characterized in solution without observing retro-[2+2] or other side-reactions. Therefore, this is a very convenient method for the synthesis of this kind of new bis-amino acid.

The characterization of compounds **2** and **3** was carried out on the basis of their MS and NMR data (Appendix A). The ESI^+^ spectra of **2** and **3** showed perfect agreement with the structures shown in Figure 5 and Figure 6. The analysis of the NMR spectra of **2** (Appendix A) enabled inference of the cyclobutane scaffold, but the absence of significative NOE cross-peaks precluded knowledge of the spatial distribution of the cyclobutane substituents, and hence, their full structural characterization. Fortunately, the NMR data of **3a** and **3b** (Appendix A) allowed an unambiguous structural determination, showing that photodimerization occurred between the exocyclic C_1’_=C_4_ bond attached to the oxazolone ring of one molecule and the styryl C_3’_=C_2’_ bond of the other molecule (Figure 5 and Figure 6). The simultaneous presence in the NMR spectra of signals due to the vinyl oxazolone proton (C_1’_(*H*)=C) and the styryl fragment (Ar-C_3’_(*H*)=C_2’_(*H*)-) supports this hypothesis. The ^1^H COSY spectrum of **3a** (Appendix A) showed a clear correlation between the vinylic protons and the peak at 3.76 ppm, which was then assigned to H-C1’. The COSY spectrum of **3a** also showed a correlation between the proton at 3.30 ppm with those appearing at 3.76 ppm and 4.68 ppm, but no correlation was observed between the latter, showing that the signal at 3.30 ppm was due to the H at C3’. The multiplicity of the signals due to the three chemically inequivalent protons at C1’, C2’ and C3’ showed that only head-to-head coupling was possible. In turn, the peak at 4.68 ppm (H at C2’) correlated with the peak at 6.50 ppm, which was then assigned to the proton at the N-C=CH group. This structural information of **3a** is presented in Figure 7a, and similar conclusions can be obtained from the analysis of the ^1^H COSY of **3b**.

To determine the relative spatial arrangement of the cyclobutane substituents, the ^1^H NOESY spectrum of **3a** (Appendix A) was measured, as well as the selective ^1^H 1D-SELNOESY spectra of **3b** (Appendix A). The ^1^H NOESY spectrum of **3a** showed clear NOE correlations between the H at 3.30 ppm (H_3’_) and the ortho H of the styryl fragment, as well as with the H at 6.50 ppm, showing that all these groups were close in space and pointing to the same side of the molecular plane defined by the cyclobutane (upwards of the plane in Figure 7b). The vinylic H at 6.50 ppm also showed an intense NOE with one of the NH protons (7.72 ppm); this indicated that the configuration of this alkene fragment was *E*. In addition, the signal at 4.68 ppm (H-C_2’_) showed strong NOEs with the H_ortho_ of the Ph ring at C_3’_ and with the NH at 8.25 ppm, allowing their full assignation and showing that all these groups were also on the same side of the molecular plane (downwards of the plane in Figure 7b). Once the structure of **3** was established and taking into account that the ring opening reaction of the oxazolone did not alter the configuration of the cyclobutane carbons [32], we assume the structures shown in Figure 5 for the cyclobutane-bis(oxazolones) **2**.

### 2.3. Characterization of the Oxazolone Reactive Excited State by Transient Absorption Spectroscopy

The remarkable difference among the cyclobutanes **2a**, **2b**, **2d**, **2f** and the truxinic derivatives **3a**, **3b** with respect to examples previously reported is that, in the cyclobutane rings **2** and **3,** the exocyclic C=C bond attached to the oxazolone has reacted with a C=C bond quite differently, instead of with another exocyclic C=C bond. This fact is of considerable interest because it opens the door for new heterodimerizations involving oxazolones and other alkenes as sources of new cyclobutane-amino acids. A key point for understanding this different reactivity is the characterization of the reactive species in the excited state; this task was accomplished using transient absorption spectroscopy.

A comparison of the absorption spectrum of [Ru(bpy)_3_](BF_4_)_2_, which showed strong absorption at 460 nm [49], with the spectra of the oxazolones **1** (having broad absorption around 380–400 nm; see Appendix A) indicated that the absorption of the incident light of the Kessil lamp (456 nm) was mostly produced by the ruthenium species. This excitation resulted in the formation of its singlet excited state ^1^[Ru(bpy)_3_^2+^]*, which was followed by fast ISC (intersystem crossing) to give its reactive triplet excited state ^3^[Ru(bpy)_3_^2+^]*. Hence, there are two main possibilities for the reaction between ^3^[Ru(bpy)_3_^2+^]* and the oxazolone: either an electron transfer (a redox process) or an energy transfer. Taking into account that the redox potentials reported for the Ru species are [Ru(bpy)_3_]^3+^/[Ru(bpy)_3_]^2+^ and [Ru(bpy)_3_]^2+^/[Ru(bpy)_3_]^+^ (+1.29 V and −1.33 V vs. SCE, respectively) [49,50,51], and that the measured redox potentials for oxazolone **1a** (Appendix A; CH_2_Cl_2_; NBu_4_PF_6_ 1M) are E^0^(oxazolone^•+^/oxazolone) = +1.65 V and E^0^(oxazolone/oxazolone^•−^) = −1.15 V, it is possible to see that both redox processes are thermodynamically unfavorable by ΔG^0^_et_ = +0.08 eV and + 0.60 eV, respectively. Therefore, we assume at this point that the reaction takes place through an energy transfer process from the ruthenium to the oxazolone, as will be shown later (Section 2.4).

The energy transfer from ^3^[Ru(bpy)_3_^2+^]* to oxazolone promoted efficient quenching of the emission of the ruthenium sensitizer when increasing amounts of oxazolone **1a** were added. As previously reported, the transient absorption spectrum of the ^3^[Ru(bpy)_3_^2+^]* (Appendix A) after laser excitation at 532 nm showed a stimulated emission at 620 nm, a ground state bleaching at 450 nm, and a transient absorption band at 360 nm [32]. The three bands showed the same kinetic behavior, confirming that all of them belonged to the same ruthenium species. The effect of the addition of increasing the amounts of oxazolone **1a** to a [Ru(bpy)_3_]^2+^ solution was to produce effective quenching of the emission, as is clearly shown in Figure 8a. Analysis of the decay using the Stern–Volmer equation produced a value of 4.25 × 10^10^ M^−1^s^−1^ for the deactivation constant, showing that the reaction was diffusion-controlled [52]. Similar values of this constant were found for the oxazolones **1b** and **1d** (Appendix A). 

Moreover, the measurement of the transient absorption spectra of deoxygenated solutions of [Ru(bpy)_3_]^2+^ in CH_2_Cl_2_ and in the presence of oxazolones **1** at different times after the laser pulse enabled visualization of the formation of a new transient species. This is shown in Figure 8b, where it is clear that the spectrum of the triplet excited state ^3^[Ru(bpy)_3_^2+^]* vanished with time, and a new species with an absorption band at 450 nm emerged and disappeared in a few microseconds. At a short interval after the pulse (12 or 44 ns, black or red lines, respectively) the main species was the triplet of the Ru, while at long intervals (552 or 924 ns, grey and ochre lines), the triplet of the Ru had almost disappeared due to energy transfer to the oxazolone and the subsequent formation of a transient species, which was visible in the emerging absorption at 450 nm. Our proposal is that this transient species is the oxazolone excited state, which appears as the photosensitizer disappears. Figure 9 shows the decay curve (black line) measured at 470 nm of the new species detected in deoxygenated solutions by the reaction of **1a** with ^3^[Ru(bpy)_3_^2+^]*. The half-life of this intermediate, determined from the exponential decay, was 12.15 μs in deoxygenated solution. Both the half-life of this intermediate, in the scale of microseconds, as well as the fast deactivation of this species in the presence of O_2_ (red line in Figure 9; a value of 1 × 10^9^ M^−1^s^−1^ was determined for the deactivation rate constant) points to the triplet nature of this excited state. A similar value for the half-life was measured for **1b** (10.72 μs), while for **1d,** the value was shorter (726.1 ns), although this still suggests a triplet nature of the excited state. From all these data, we can conclude that the ruthenium species behaves as a photosensitizer of the oxazolones **1**, whose reactive state ^3^[oxa-**1**]* is a triplet excited state generated, presumably, by an energy transfer process from ^3^[Ru(bpy)_3_^2+^]* to the oxazolone oxa-**1**, as shown in Figure 10. Due to the importance of the correct characterization of the reactive species, it was further studied using DFT methods.

### 2.4. Characterization of the Reactive Excited State by DFT Methods

We employed DFT (ωB97X-D/def2-QZVPP//ωB97X-D/6-31+G(d), SMD = dichloromethane in all calculations) [53,54,55,56,57,58,59,60,61] to gain insights regarding the T_1_ excited state of **1a**, ^3^[oxa-**1a**]*, which could reveal the reasons behind the reactivity differences shown by the allyliden-5(4*H*)-oxazolone **1a** and aryliden-5(4*H*)-oxazolones **4** (Figure 11). In this study, we focused on the major isomer *E,Z*-**1a** observed experimentally.

There were significant variations in the excited state properties of the two oxazolone derivatives considered. The Gibbs free energy (G) gap was considerably lower in **1a** (29.8 vs. 35.7 kcal·mol^−1^), which was probably due to the higher delocalization of the two unpaired electrons from T_1_ of **1a** with the longer conjugated system (Figure 11A). Moreover, the atoms with higher spin density differed: in **1a,** the α and δ C atoms of the diene system showed the highest spin density (0.58 and 0.35 unpaired electrons), while in **4,** the α and β positions were predominant (0.70 and 0.42 unpaired electrons). The unpaired electrons accumulated in the reactive positions in each species, explaining the switch in reactivity observed.

The T_1_ geometries of the two oxazolones also differed significantly. While in **1a,** the conjugated system was planar (θ = 178°), the Ph ring was perpendicular to the oxazolone ring in **4** (θ = 96°) (Figure 11A). Interestingly, we found that **1a** and **4** showed rotation transition states (TSs) to interconvert from *E* to *Z* structures with low barriers (2.6 and 2.5 kcal·mol^−1^, respectively, Appendix A). The low calculated barriers were consistent with previous photophysical experimental results, which indicated that the same triplet state was rapidly formed when starting from the *E* and *Z* forms of compound 4 [32]. These rapid isomerization processes may have been an important contributing factor to the low yields obtained experimentally since multiple competitive isomers of 2 can be formed. We studied how these T_1_ geometries distorted over time using molecular dynamics (MD, Figure 11B) [62]. The population distribution of **1a** was relatively narrow and rested around the planar system (average θ = 168°, approximate range of θ = 130-180°). This result suggested that the perpendicular rotation TS (θ = 90°) with low energy was not immediately reached. In contrast, the geometry of **4** can easily switch from the planar to the perpendicular geometry (average θ = 110°, approximate range of θ = 60–180°).
Figure 11(**A**) G difference between optimized geometries of **1a** and **4** in S_0_ and T_1_, along with natural spin populations [63] of their conjugated systems and representations of T_1_ geometries. (**B**) Distribution of the θ dihedral angle from MD simulations of **1a** and **4** in T_1_. Since rotations were symmetrical in both directions, any values above 180° were converted to their equivalent degrees between 0 and 180° for clarity (i.e., 190° would be equivalent to 170°).
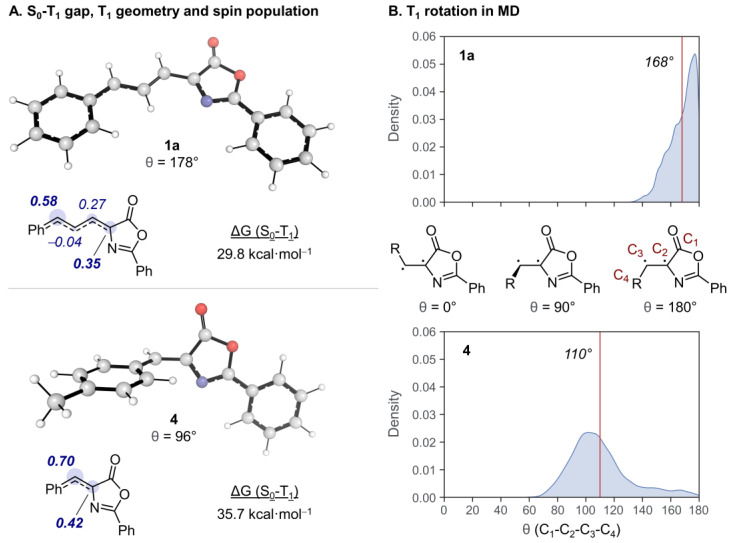



## 3. Materials and Methods

### 3.1. General Procedures

The [2+2] photocycloadditions were performed under Ar atmosphere, using deoxygenated CH_2_Cl_2_. Other reactions were carried out in reagent-grade solvents in open air. Flash column liquid chromatographies were carried out on silica gel (70−230 μm). ^1^H, ^13^C, and ^19^F NMR spectra were measured in CDCl_3_, CD_2_Cl_2_, or toluene-d_8_ solutions at 25 °C on Bruker AV300 or Bruker AV500 spectrometers (δ in ppm, J in Hz) at ^1^H operating frequencies of 300.13 MHz and 500.13 MHz, respectively. ^1^H and ^13^C NMR spectra were referenced using the solvent signal as the internal standard, while ^19^F NMR spectra were referenced to CFCl_3_. The assignment of ^1^H NMR peaks was performed with the help of 2D ^1^H−COSY, 2D ^1^H-NOESY and 1D ^1^H SELNOE experiments (mixing times of 1.2−1.8 s, as a function of the irradiated signal), while ^13^C NMR peaks were identified using ^1^H−^13^C edited HSQC and ^1^H−^13^C HMBC 2D experiments. HRMS and ESI (ESI+) mass spectra were recorded using a MicroToF Q, API-Q-ToF ESI instrument with a mass range from m/z 20 to 3000 and a mass resolution of 15,000 (full width at half-maximum). The absorption spectra of **1a−1h** were measured in an Evolution 600 UV−Vis spectrophotometer in CH_2_Cl_2_ solutions (10^−5^ M), the excitation and emission spectra were recorded on a Horiba Jobin Yvon Fluoromax-P spectrophotometer, and the quenching of the phosphorescence of [Ru(bpy)_3_](BF_4_)_2_ by **1a** was measured on a Horiba Jobin Yvon Fluorolog FL-3.11 spectrophotometer. All spectra were recorded at 25 °C using 10 mm quartz cuvettes. The cyclic voltammetry of **1a** (5 · 10^−4^ M) was carried out using a Voltalab50 potentiostat/galvanostat, equipped with a glass electrochemical cell with the typical configuration of three electrodes: a Pt working electrode, another Pt counter electrode, and the SCE electrode. The solution of the pure electrolyte (NBu_4_PF_6_, 0.1 M) was measured over the whole window of the solvent (CH_2_Cl_2_) to check the absence of electroactive impurities. The oxazolones **1a**–**1h** were prepared following published methods [33,34,35,36,37,38,39,40]. The photosensitizer [Ru(bpy)_3_](BF_4_)_2_ was synthesized following methods found in [64,65] and stored under Ar at 4 °C. The melting points (degrees Celsius) were determined on a Gallenkamp apparatus and were uncorrected.

### 3.2. Irradiation Setup

The solution of oxazolone **1a–1h** in a Schlenck flask was irradiated at 456 nm by a Kessil PR160L LED lamp (maximal power of 50W, but this intensity can be tuned). The lamp and the flask were placed 5 cm apart to avoid overheating the solution. To maximize the light of the lamp received by the solution, a mirror was placed in front of the lamp.

### 3.3. X-ray Crystallography

The crystals of oxazolone **1a** were obtained by slow evaporation at room temperature of a CDCl_3_ solution of **1a**. The crystals were assembled at low temperature (100 K) using a specific commercial system called MiTeGen micromunts Cryoloop. Diffraction data were obtained on a Bruker D8 Venture diffractometer using Mo-Kα radiation (λ = 0.71073 Å), filtered through a graphite monochromator and multilayer optics, at low temperature (100 K). The diffraction frames were integrated using SAINT [66] and the integrated intensities were corrected for absorption with SADABS [67]. The structure was determined and developed by Fourier methods [68]. All non-hydrogen atoms were refined with anisotropic thermal parameters. The H atoms were placed at idealized positions and treated as riding atoms. Both structure resolution and structure refinement were carried out using a commercial Bruker package (Bruker APEX3 software package) [69]. CCDC-1972175 contains the supplementary crystallographic data for this paper. These data can be obtained free of charge from The Cambridge Crystallographic Data Centre via www.ccdc.cam.ac.uk/data_request/cif (accessed on 9 December 2020).

### 3.4. Photophysical Experiments

The laser flash photolysis (LFP) experiments were performed using a pulsed Nd:YAG SL404G-10 Spectron Laser Systems laser at an excitation wavelength of 532 nm. The energy of the single pulses (around 10 ns duration) was smaller than 15 mJ pulse^−1^. The LFP system is formed by the following elements: a pulsed laser, a pulsed Lo255 Oriel Xe-lamp, an Oriel monochromator model 77,200, an Oriel photomultiplier tube (PMT) housing, a power supply model 70,705 PMT, and a Tektronix oscilloscope TDS-640A. Quenching rate constants (*k*_q_) were determined according to the Stern−Volmer equation 1/τ = 1/τ_ο_ + *k*_q_[Q]. In this equation τ_ο_ is the triplet lifetime of Ru(bpy)_3_^2+^ in the absence of oxazolone (Q), τ is the lifetime of ^3^[Ru(bpy)_3_^2+^]* in the presence of a given concentration of oxazolone, and [Q] is the oxazolone concentration. The quenching rate constants (*k*_q_, M^−1^ s^−1^) were the corresponding slopes of the linear fittings of the Stern−Volmer plots.

### 3.5. General Procedure for the Synthesis of 2-Aryl-4-(E-3′-aryl-allylidene)-5(4H)-oxazolones ***1a***–***1h***

The synthesis of **1a**–**1h** was carried out following published procedures [33,34,35,36,37,38,39,40], which are detailed here for **1a**. All other oxazolones were prepared using the same procedure.

#### 3.5.1. Synthesis of 2-Phenyl-4-(E-3′-phenylallylidene)-5(4H)-oxazolone **1a** (EZ and EE Isomers)

A Schlenk flask was charged with cinnamaldehyde (0.776 g, 5.88 mmol), hippuric acid (1.054 g, 5.88 mmol), sodium acetate (0.467 g, 5.69 mmol), and acetic anhydride (4 mL). This mixture was stirred while heating at 100 °C for 2 h. Then, the resulting suspension was allowed to cool, giving oxazolone **1a** as a viscous waxy solid. This solid was vigorously stirred with 25 mL of ethanol. The solid thus obtained was filtered, washed with additional ethanol (60 mL), dried by suction, and characterized as the mixture of *EZ* and *EE* isomers of oxazolone **1a**. Obtained: 1.002 g (62% yield). Ratio 2.5/1 (*EZ*/*EE*). HRMS (ESI^+^) [*m*/*z*]: calculated for [C_18_H_13_NO_2_Na]^+^ = 298.0844; found 298.0833. ^1^H NMR major isomer (*EZ*) (toluene-d8, 300.13 MHz, 25 °C): δ = 7.99 (dm, 2H, H_o_, NCOPh, ^3^J_HH_ = 7.7 Hz), 7.66 (dd, 1H, H_2’_, ^3^J_HH3_ = 15.7 Hz, ^3^J_HH1_ = 11.6 Hz), 7.17 (m, 2H, H_o_, Ph), 7.10–6.95 (m, 6H, H_m_ + H_p_, NCOPh + H_m_ + H_p,_ Ph), 6.88 (dd, 1H, H_1’_, ^3^J_HH_ = 11.6 Hz, ^4^J_HH_ = 0.8 Hz), 6.51 (d, 1H, H_3’,_
^3^J_HH_ = 15.7 Hz). ^13^C{^1^H} NMR major isomer (*EZ*) (toluene-d8, 300.13 MHz, 25 °C): δ = 166.12 (CO), 162.62 (CN), 143.03 (C_3’_), 132.74 (C_p_, NCOPh), 132.37 (C_1’_), 136.53, 134.76, 126.49 (C_i_, NCOPh + C_i_, Ph + C_q_, oxa, C_4_), 129.67(C_p_, Ph), 128.97, 128.85 (C_m_, Ph + C_m_, NCOPh), 128.33 (C_o,_ Ph), 128.04 (C_o,_ NCOPh), 123.74 (C_2’_).

#### 3.5.2. Synthesis of 2-Phenyl-4-(E-3′-(4-chlorophenyl)allylidene)-5(4H)-oxazolone **1b** (EZ and EE Isomers)

**1b**, yellow solid. 4-chlorocinnamaldehyde (1.042 g, 6.28 mmol), hippuric acid (1.084 g, 6.05 mmol), sodium acetate (0.528 g, 6.43 mmol) and acetic anhydride (2 mL). Obtained: 1.368 g (74% yield). Ratio 1.7/1 (*EZ*/*EE*). HRMS (ESI^+^) [*m*/*z*]: calculated for [C_18_H_12_ClNO_2_Na]^+^ = 332.0454; found 332.0441. ^1^H NMR major isomer (*EZ*) (CDCl_3_, 500.13 MHz, 25 °C): δ = 8.14 (m, 2H, H_o_, NCOPh), 7.65 (dd, 1H, H_2’_, ^3^J_HH3_ = 15.6 Hz, ^3^J_HH1_ = 11.6 Hz), 7.61–7.49 (m, 5H, H_m_, H_p_, NCOPh + H_o_, C_6_H_4_), 7.37 (m, 2H, H_m_, C_6_H_4_), 7.11 (dd, 1H, H_1’_, ^3^J_HH_ = 11.6 Hz, ^4^J_HH_ = 0.8 Hz), 7.06 (d broad, 1H, H_3’_, ^3^J_HH_ = 15.6 Hz). ^13^C{^1^H} NMR major isomer (*EZ*) (CDCl_3_, 125.7 MHz, 25 °C): δ = 166.68 (CO), 162.45 (CN), 135.77, 125.62, 125.61 (C_i_, C_6_H_4_ + C_i_, NCOPh + C_p_, C_6_H_4_), 142.12 (C_3’_), 134.48 (C_q_, oxa, C_4_), 132.32 (C_1’_), 133.22, 129.24, 129.04, 128.98 (C_m_, C_p_, NCOPh + C_o_, C_m_, C_6_H_4_), 128.20 (C_o_, NCOPh), 123.88 (C_2’_).

#### 3.5.3. Synthesis of 2-Phenyl-4-(E-3′-(2-nitrophenyl)allylidene)-5(4H)-oxazolone **1c** (EZ and EE Isomers)

**1c**, yellow solid. 2-nitrocinnamaldehyde (1.122 g, 6.33 mmol), hippuric acid (1.128 g, 6.30 mmol), sodium acetate (0.556 g, 6.77 mmol) and acetic anhydride (2 mL). Obtained: 1.865 g (92% yield). Ratio 1.7/1 (*EZ*/*EE*). HRMS (ESI^+^) [*m*/*z*]: calculated for [C_18_H_12_N_2_O_4_Na]^+^ = 343.0695; found 343.0696. ^1^H NMR major isomer (*EZ*) (toluene-d8, 500.13 MHz, 25 °C): δ = 7.96 (m, 2H, H_o_, NCOPh, ^3^J_HH_ = 7.0 Hz), 7.46 (dd, 1H, H_2’_, ^3^J_HH3_ = 15.6 Hz, ^3^J_HH1_ = 11.4 Hz), 7.39 (m, 1H, H_3_, C_6_H_4_), 7.10–6.95 (5H, H_m_, H_p_, NCOPh, H_6,_ C_6_H_4_, H_1’_), 6.78 (m, 1H, H_4_, C_6_H_4_), 6.74 (dd, 1H, H_3’_, ^3^J_HH2_ = 11.4 Hz, ^4^J_HH3_ = 1.0 Hz), 6.64 (m, 1H, H_5_, C_6_H_4_). ^13^C{^1^H} NMR major isomer (*EZ*) (CDCl_3_, 125.7 MHz, 25 °C): δ = 166.35 (CO), 163.34 (CN), 148.31 (C_q_-NO_2_, C_6_H_4_), 137.19, 133.61, 133.38, 129.86, 129.12, 128.72, 128.46, 127.86 (C_m_, C_p_, NCOPh + C_2_, C_3_, C_4_, C_5_, C_6_, C_6_H_4_), 136.09, 131.52, 125.51 (C_i_, C_6_H_4_ + C_i_, NCOPh + C_q_, Oxazolone, C_4_), 131.31 (C_1_), 125.19 (C_3_, Ph).

#### 3.5.4. Synthesis of (Z)-2-Phenyl-4-(E-2′-methyl-3′-phenylallyliden)-5(4H)-oxazolone **1d**

**1d**, yellow solid. α-methyl-cinnamaldehyde (1.93 mL, 13.82 mmol), hippuric acid (2.485 g, 13.87 mmol), sodium acetate (1.355 g, 16.51 mmol) and acetic anhydride (4 mL). Obtained: 3.484 g (87% yield). HRMS (ESI^+^) [*m*/*z*]: calculated for [C_19_H_15_NO_2_Na]^+^ = 312.1000; found 312.0991. ^1^H NMR (CDCl_3_, 500.13 MHz, 25 °C): δ = 8.12 (dm, 2H, H_o_, NCOPh, ^3^J_HH_ = 7.0 Hz), 7.59 (tt, 1H, H_p_, NCOPh, ^3^J_HH_ = 7.0 Hz, ^4^J_HH_ = 2.2 Hz), 7.51 (m, 2H, H_m_, NCOPh), 7.46 (m, 2H, H_o_, Ph), 7.42 (m, 2H, H_m_, Ph), 7.34 (tt, 1H, H_p_, Ph, ^3^J_HH_ = 7.0 Hz, ^4^J_HH_ = 2.2 Hz), 7.16 (s broad, 1H, H_3’_), 7.07 (d, 1H, H_1’_, ^4^J_HH_ = 1.0 Hz), 2.59 (d, 3H, 2-Me, ^4^J_HH_ = 1.0 Hz). ^13^C{^1^H} NMR (CDCl_3_, 125.7 MHz, 25 °C): δ = 168.32 (CO), 161.86 (CN), 143.93 (C_3’_), 137.74 (C_1’_), 136.64 (C_i_, Ph), 135.83 (C_q_, oxazolone, C_4_), 133.09 (C_p_, NCOPh), 132.67 (C_2’_), 130.05 (C_o,_ Ph), 129.03 (C_m_, Ph), 128.62 (2C overlapped, C_m_, NCOPh + C_p_-Ph), 128.21 (C_o,_ NCOPh), 126.01 (C_i_, NCOPh), 16.81 (Me).

#### 3.5.5. Synthesis of (Z)-2-Phenyl-4-(Z-2′-bromo-3′-phenylallylidene)-5(4H)-oxazolone **1e**

**1e**, yellow solid. α-bromo-cinnamaldehyde (2.461 g, 11.72 mmol), hippuric acid (2.110 g, 11.78 mmol), sodium acetate (0.962 g, 11.69 mmol) and acetic anhydride (4 mL). Obtained: 1.282 g (32% yield). HRMS (ESI^+^) [*m*/*z*]: calculated for [C_18_H_12_BrNO_2_Na]^+^ = 375.9949; found 375.9954. ^1^H NMR (CDCl_3_, 500.13 MHz, 25 °C): δ = 8.44 (s, 1H, H_3’_), 8.15 (m, 2H, H_o_, NCOPh, ^3^J_HH_ = 7.2 Hz), 7.89 (m, 2H, H_o_, Ph, ^3^J_HH_ = 7.0 Hz), 7.63 (tt, 1H, H_p_, NCOPh, ^3^J_HH_ = 7.4 Hz, ^4^J_HH_ = 1.1 Hz), 7.53 (m, 2H, H_m_, NCOPh), 7.47–7.41 (m, 3H, H_p_ + H_m_, Ph), 7.08 (s, 1H, H_1’_).^13^C{^1^H} NMR (CDCl_3_, 125.7 MHz, 25 °C): δ = 166.59 (CO), 164.81 (CN), 141.86 (C_3’_), 135.30 (C_i_, Ph), 133.86 (C_p_, NCOPh), 133.38 (C_q_, oxazolone, C_4_), 132.59 (C_1’_), 130.12 (C_o_, Ph), 129.93 (C_p_, Ph), 129.06 (C_m_, NCOPh), 128.62 (C_m_, Ph), 128.43 (C_o_, NCOPh), 125.20 (C_i_, NCOPh), 114.75 (C_2’_).

#### 3.5.6. Synthesis of 2-(4-Cyanophenyl)-4-(E-3′-phenylallylidene)-5(4H)-oxazolone **1f** (EZ and EE Isomers)

**1f**, yellow solid. Cinnamaldehyde (2.100 g, 15.89 mmol), 4-cyanohippuric acid (3.089 g, 15.12 mmol), sodium acetate (1.381 g, 16.82 mmol) and acetic anhydride (4.5 mL). Obtained: 1.990 g (65% yield). Ratio 3.3/1 (*EZ*/*EE*). HRMS (ESI^+^) [*m*/*z*]: calculated for [C_19_H_13_N_2_O_2_]^+^ = [M + H]^+^ = 301.0977; found 301.0960. ^1^H NMR major isomer (*EZ*) (CDCl_3_, 300.13 MHz, 25 °C): δ = 8.23 (dm, 2H, H_o_, NCOC_6_H_4_CN), 7.80 (tm, 2H, H_m_, NCOC_6_H_4_CN), 7.68 (dd, 1H, H_2’_, ^3^J_H2H3_ = 15.7 Hz, ^3^J_H2H1_ = 11.6 Hz), 7.62 (m, 2H, H_o_, Ph), 7.44–7.38 (m, 3H, H_m_ + H_p_, Ph), 7.23 (dd, 1H, H_1’_, ^3^J_H1H2_ = 11.6 Hz, ^4^J_H1H3_ = 0.9 Hz), 7.19 (d broad, 1H, H_3’_, ^3^J_H3H2_ = 15.6 Hz). ^13^C{^1^H} NMR Major isomer (*EZ*) (CDCl_3_, 75.5 MHz, 25 °C): δ = 166.08 (CO), 160.41 (CN), 145.63 (C_3’_), 135.85 (C_i_, Ph), 135.32 (C_1’_), 133.54 (C_q_, oxazolone, C_4_), 132.75 (C_m_, NCOC_6_H_4_), 130.57 (C_p_, Ph), 129.86 (CN), 129.19 (C_m_, Ph), 128.53 (C_o_, NCOC_6_H_4_), 128.32 (C_o_, Ph), 123.30 (C_2’_), 118.05 (C_i_, NCOC_6_H_4_), 116.21 (C_p_, NCOC_6_H_4_).

#### 3.5.7. Synthesis of 2-(E-Styryl)-4-(E-3′-phenylallylidene)-5(4H)-oxazolone **1g** (EZ and EE Isomers)

**1g**, yellow solid. Cinnamaldehyde (0.735 g, 5.56 mmol), cinnamoylglycine (1.063 g, 5.18 mmol), sodium acetate (0.4222 g, 5.14 mmol) and acetic anhydride (1.5 mL). Obtained: 0.180 g (17% yield). Ratio 1.7/1 (*EZ*/*EE*). HRMS (ESI^+^) [*m*/*z*]: calculated for [C_20_H_15_NO_2_]^+^ = 301.1103; found 301.1164. ^1^H NMR major isomer (*EZ*) (CDCl_3_, 300.13 MHz, 25 °C): δ = 8.10 (dd, 1H, H_2’_, ^3^J_HH3_ = 15.6 Hz, ^3^J_HH1_ = 12.1 Hz), 7.70–7.35 (m, 11H, H_o_, H_m_, H_p,_ NCOPh + H_o_, H_m_, H_p_, Ph + = CH_5”_), 7.20 (dd, 1H, H_1’_, ^3^J_HH_ = 12.1 Hz, ^4^J_HH_ = 0.7 Hz), 7.08 (d broad, 1H, H_3’_, ^3^J_HH_ = 15.6 Hz), 6.72 (d, 1H, =CH_4”_, ^3^J_HH_ = 16.2 Hz). ^13^C{^1^H} NMR major isomer (*EZ*) (CDCl_3_, 75.5 MHz, 25 °C): δ = 166.33 (CO), 161.69 (CN), 144.91 (C_3’_), 142.70 (C_5”_), 137.24 (C_1’_), 135.99 (C_i_, Ph), 134.69 (C_i_, NCOPh), 133.82 (C_q_, oxazolone, C_4_), 130.57, 129.99, 129.08, 128.95, 128.04 (2C overlapped) (C_o_, C_m_, C_p_, Ph + C_o_, C_m_, C_p_, NCOPh), 123.45 (C_2’_), 113.17 (C_4”_).

#### 3.5.8. Synthesis of (Z)-2-Phenyl-4-(3’,3′-diphenylallylidene)-5(4H)-oxazolone **1h**

**1h**, yellow solid. β-phenyl-cinnamaldehyde (0.502 g, 2.41 mmol), hippuric acid (0.432 g, 2.41 mmol), sodium acetate (0.202 g, 2.46 mmol) and acetic anhydride (1 mL). Obtained: 0.341 g (40% yield). HRMS (ESI^+^) [*m*/*z*]: calculated for [C_24_H_17_NO_2_Na]^+^ = 374.1157; found 374.1124. ^1^H NMR (CDCl_3_, 300.13 MHz, 25 °C): δ = 8.14 (m, 2H, H_o_, NCOPh, ^3^J_HH_ = 7.0 Hz), 7.68 (d, 1H, H_2’_, ^3^J_HH_ = 12.1 Hz), 7.59 (m, 1H, H_p,_ NCOPh), 7.53 (m, 2H, H_m_, NCOPh), 7.49–7.26 (m, 10H, Ph), 7.08 (d, 1H, H_1’_, ^3^J_HH_ = 12.1 Hz). ^13^C{^1^H} NMR (CDCl_3_, 75.5 MHz, 25 °C): δ = 167.02 (CO), 162.08 (CN), 154.92 (C_3’_), 141.15, 138.36 (C_i_, Ph), 134.74 (C_q_, oxazolone, C_4_), 133.16 (C_p_, NCOPh), 131.09 (C_1’_), 130.78, 129.62, 129.06 (2C overlaped), 128.86, 128.67, 128.60 (C_m_, NCOPh + C_o_, C_m_, C_p_, 2 Ph), 128.25 (C_o_, NCOPh), 125.89 (C_i_, NCOPh), 122.52 (C_2’_).

### 3.6. Synthesis and Characterization of Cyclobutane-bis(oxazolone) Intermediates ***2a***, ***2b***, ***2d*** and ***2f***

The synthesis of cyclobutanes **2** was carried out by [2+2]-photocycloaddition of oxazolones **1**, photosensitized by [Ru(bpy)_3_](BF_4_)_2_. This procedure is detailed here for oxazolone **1a**. All other cyclobutanes were prepared using the same procedure.

#### 3.6.1. Synthesis of Cyclobutane **2a**

The oxazolone **1a** (0.138 g, 0.502 mmol) and the photocatalyst [Ru(bpy)_3_](BF_4_)_2_ (0.0186 g, 0.025 mmol) were dissolved in deoxygenated CH_2_Cl_2_ (5 mL) under Ar atmosphere. This red solution was irradiated with the blue light (456 nm) provided by a Kessil LED lamp (50 W) for 18 h. Then the solvent was evaporated to dryness and the residue characterized as a mixture of isomers of cyclobutane **2a** (10.1/1 molar ratio): The major isomer in this mixture was separated and purified by flash chromatography on silica gel using hexane/ethyl acetate as eluent (8/2 ratio). The orange band collected was evaporated to dryness to give cyclobutane **2a** as an orange-yellowish solid. Obtained: 0.060 g (43% yield). HRMS (ESI^+^) [*m*/*z*]: calculated for [C_36_H_26_N_2_NaO_4_]^+^ = 573.1793 [M+Na]^+^; found: 573.1778. ^1^H NMR (CDCl_3_, 300.13 MHz, 25 °C): δ = 7.97 (m, 2H, H_o_, Ph-oxa), 7.82 (m, 2H, H_o_, Ph-oxa), 7.50 (m, 2H, H_p_, Ph-oxa), 7.39 (m, 2H, H_m_, Ph-oxa), 7.23–7.15 (m, 6H, 4H_o_ + 2H_m_, Ph), 7.11–6.98 (m, 6H, 4H_m_ + 2H_p_, Ph), 6.26 (dd, 1H, H_2′-_vinyl, ^3^J_HH_ = 10 Hz, ^4^J_HH_ = 1.8 Hz), 6.16 (d, 1H, =CH_1′-_oxa, ^3^J_HH_ = 12 Hz), 5.69 (dd, 1H, H_3′-_vinyl, ^3^J_HH_ = 10 Hz, ^4^J_HH_ = 1.8 Hz), 4.97 (m, 1H, H_2’_, cyclo), 3.78–3.69 (m, 2H, H_3’_ + H_1’_, cyclo). ^13^C{^1^H} NMR (CDCl_3_, 75.5 MHz, 25 °C): δ = 177.10 (C(O)O-cyclo), 164.64 (C(O)O-vinyl), 162.46, 161.93 (C=N), 141.68, 140.25 (C_i_, Ph), 138.69 (C_q_, oxa), 137.48 (CH vinyl, C_2’_), 138.86 (C_1’_H-oxa), 133.44, 133.15 (C_p_, Ph-oxa), 128.95, 128.92, 128.63, 128.55, 128.46, 128.21, 128.18 (C_m_, C_o_, Ph + Ph-oxa), 127.25, 126.96 (C_p_, Ph), 125.39, 125.03 (C_i_, Ph-oxa), 123.45 (CH vinyl, C_3’_), 73.42 (C_4_, cycle), 50.45 (CH cyclo, C_3’_), 49.03 (CH cyclo, C_1’_), 44.06 (CH cyclo, C_2’_).

#### 3.6.2. Synthesis of Cyclobutane **2b**

The synthesis of the cyclobutane **2b** was carried out following the same experimental procedure described for **2a**. Therefore, **1b** (0.154 g, 0.498 mmol) and the photocatalyst (0.0186 g, 0.025 mmol) were irradiated in CH_2_Cl_2_ (5 mL) with blue light (456 nm) for 18 h to give a mixture of two cyclobutanes in 3.6/1 molar ratio, which could not be separated by column chromatography. The major isomer in this mixture was characterized spectroscopically as **2b**. Obtained: 0.030 g (19% yield). HRMS (ESI^+^) [*m*/*z*]: calculated for [C_36_H_24_Cl_2_N_2_NaO_4_]^+^ = 641.1013 [M+Na]^+^; found: 641.0977. ^1^H NMR (CDCl_3_, 300.13 MHz, 25 °C): δ = 7.95 (m, 2H, H_o_, Ph-oxa), 7.83 (m, 2H, H_o_, Ph-oxa), 7.56–7.51 (m, 4H, H_p_, Ph-oxa + H_m_, Ph-Cl), 7.44–7.38 (m, 4H, H_m_, Ph-oxa), 7.19 (m, 2H, H_o_, Ph-Cl), 7.10 (m, 2H, H_m_, Ph-Cl), 6.93 (m, 2H, H_o_, Ph-Cl), 6.19 (dd, 1H, H_2′-_ vinyl, ^3^J_HH_ = 9.8 Hz, ^4^J_HH_ = 1.9 Hz), 6.10 (d, 1H, =CH_1′-_oxa, ^3^J_HH_ = 12 Hz), 5.72 (dd, 1H, H_3′-_vinyl, ^3^J_HH_ = 9.8 Hz, ^4^J_HH_ = 2.3 Hz), 4.94 (m, 1H, H_2’_, cyclo), 3.64–3.69 (m, 2H, H_3’_ + H_1’_, cyclo). ^13^C{^1^H} NMR (CDCl_3_, 75.5 MHz, 25 °C): δ = 177.05 (C(O)O-cyclo), 164.68 (C(O)O-vinyl), 162.84, 162.12 (C=N), 139.89 (C_i_, Ph-vinyl), 139.05 (C_q_, oxa), 138.58 (C_i_, Ph-vinyl), 136.52 (CH vinylic, C_2’_), 135.82 (CH-oxa), 133.63, 133.28 (C_p_, Ph-oxa), 133.16, 133.00 (C_4_, Ph-Cl), 129.90, 129.01, 128.98, 128.96, 128.81, 128.30, 128.24 (C_m_, C_o_, Ph-Cl + Ph-oxa), 125.25, 124.90 (C_i_, Ph-oxa), 124.11 (CH vinylic, C_3’_), 73.24 (C_4_, cyclo), 49.94 (CH cyclo, C_3’_), 48.47 (CH cyclo, C_1’_), 43.74 (CH cyclo, C_2’_).

#### 3.6.3. Synthesis of Cyclobutane **2d**

The synthesis of the cyclobutane **2d** was carried out following the same experimental procedure described for **2a**. Therefore, **1d** (0.154 g, 0.533 mmol) and the photocatalyst (0.0186 mg, 0.025 mmol) were irradiated in CH_2_Cl_2_ (5 mL) with blue light (456 nm) for 17 h to give a mixture of two cyclobutanes in 2.9/1 molar ratio, which could not be separated by column chromatography. The major isomer in this mixture was characterized spectroscopically as **2d**. Obtained: 0.035 g (22% yield). HRMS (ESI^+^) [*m*/*z*]: calculated for [C_38_H_30_N_2_NaO_4_]^+^ = 601.2106 [M+Na]^+^; found: 601.2081. ^1^H NMR (CDCl_3_, 300.13 MHz, 25 °C): δ = 8.00 (m, 2H, H_o_, Ph-oxa), 7.84 (m, 2H, H_o_, Ph-oxa), 7.62–7.04 (overlapped aromatics, 16H), 5.40 (s, 1H, H_3′-_vinyl), 4.06 (m, 1H, H_3’_, cyclo), 3.80 (m, 1H, H_1’_, cyclo), 1.62 (s, 3H, CH_3_-C_2’_ vinyl), 1.30 (s, 3H, CH_3_-C_2’_ cyclo). ^13^C{^1^H} NMR (CDCl_3_, 75.5 MHz, 25 °C): δ = 177.68 (C(O)O-cyclo), 167.55 (C(O)O-vinyl), 162.68, 161.73 (C=N), 143.87 (C vinylic, C_1_), 141.51 (C_i_, Ph-vinyl), 138.38 (C_i_, Ph-vinyl), 138.20 (C_q_, oxa), 137.20 (CH-oxa), 133.29, 133.04, 127.03, 126.64 (C_p_, Ph-oxa + Ph), 128.88 (br), 128.84 (br), 128.40 (br), 128.32, 127.96 (C_m_, C_o_, Ph + C_m_, Ph-oxa), 125.58, 125.69 (C_i_, Ph-oxa), 119.83 (CH vinyl, C_2’_), 77.40 (C_4_, cyclo), 54.39 (CH cyclo, C_3’_), 51.50 (CH cyclo, C_1’_), 48.51 (C cyclo, C_2’_), 22.30 (CH_3_-C vinyl), 19.92 (CH_3_-C cyclo).

#### 3.6.4. Synthesis of Cyclobutane **2f**

The synthesis of the cyclobutane **2f** was carried out following the same experimental procedure described for **2a**. Therefore, the oxazolone **1f** (0.145 g, 0.482 mmol) and the photocatalyst (0.0186 g, 0.025 mmol) were irradiated in CH_2_Cl_2_ (5 mL) with blue light (456 nm) for 18 h to give cyclobutane **2f** as a yellow solid after separation and purification by column chromatography (silica gel, hexane/ethyl acetate = 8/2 as eluent). Obtained: 0.026 g (18% yield). HRMS (ESI^+^) [*m*/*z*]: calculated for [C_38_H_24_N_4_NaO_4_]^+^ = 623.1698 [M+Na]^+^; found: 623.1656. ^1^H NMR (CDCl_3_, 300.13 MHz, 25 °C): δ = 8.16 (m, 2H, H_o_, Ph-CN), 8.03 (m, 2H, H_o_, Ph-CN), 7.83 (m, 2H, H_m_, Ph-CN), 7.65 (m, 2H, H_m_, Ph-CN), 7.42–7.31 (m, 4H, H_o_, Ph), 7.21–7.18 (m, 2H, H_m_, Ph), 7.04–6.97 (m, 4H, H_p_, Ph + H_m_, Ph), 6.33 (dd, 1H, H_2′-_vinyl, ^3^J_HH_ = 10 Hz, ^4^J_HH_ = 1.7 Hz), 6.09 (d, 1H, =CH_1′-_oxa, ^3^J_HH_ = 11 Hz), 5.68 (dd, 1H, H_3′-_vinyl, ^3^J_HH_ = 10 Hz, ^4^J_HH_ = 1.7 Hz), 4.52 (m, 1H, H_2’_, cyclo), 3.79–3.70 (m, 2H, H_3’_ + H_1’_, cyclo). ^13^C{^1^H} NMR (CDCl_3_, 75.5 MHz, 25 °C): δ = 175.95 (C(O)O-cyclo), 163.82 (C(O)O-vinyl), 162.38, 160.59 (C=N), 141.15 (C_i_, Ph-vinyl), 140.34 (C_q_, oxa), 139.32 (C_i_, Ph-cyclo), 138.20 (CH vinyl, C_2’_), 134.27 (CH-oxa), 132.80, 132.67 (C_o_, Ph), 129.18, 129.15 (CN-Ph), 128.83, 128.68, 128.58, 128.54 (br), 128.42 (C_m_, C_o_, Ph-CN + C_m_, Ph), 127.36, 127.19 (C_p_, Ph), 122.50 (CH vinylic, C_3’_), 117.80, 117.68 (C_i_, Ph), 117.00, 116.71 (C_4_-CN, Ph-CN) 73.37 (C_4_, cyclo), 50.49 (CH cyclo, C_3’_), 48.40 (CH cyclo, C_1’_), 46.39 (CH cyclo, C_2’_).

### 3.7. Synthesis and Characterization of Cyclobutanes ***3a*** and ***3b***

#### 3.7.1. Synthesis of Methyl-1-benzamido-2-((E)-2-benzamido-2-methoxycarbonylprop-1-en-1-yl)-3-phenyl-4-((E)-styryl)cyclobutane-1-carboxylate **3a**

Oxazolone **1a** (0.2982 g, 1.084 mmol) and [Ru(bpy)_3_](BF_4_)_2_ (0.0391 g, 0.053 mmol) (5% mol) were dissolved in deoxygenated CH_2_Cl_2_ under Ar atmosphere and irradiated for 18 h (456 nm, Kessil lamp, 50 W). Then the solution was evaporated to dryness (irradiation was maintained during evaporation to prevent thermal retro-[2+2]) and the oily residue treated with methanol (10 mL) and NaOMe (9 mg, 0.167 mmol). The resulting suspension was heated in an oil bath to the reflux temperature for 45 min. The resulting solution was cooled and evaporated to dryness. The residue was taken in the minimal amount of CHCl_3_ (2 mL) and subjected to flash chromatography on silica gel using CHCl_3_ as eluent. The colorless fraction containing **3a** was evaporated to dryness and treated with *n*-pentane (15 mL), giving **3a** as a white solid. The white solid was crystallized in CH_2_Cl_2_/*n*-pentane at −18 °C to give **3a** as a white crystalline solid. Obtained: 0.2401 g (80% yield). Mp: 145–146 °C. HRMS (ESI^+^) [*m*/*z*]: calculated for [C_38_H_34_N_2_NaO_6_]^+^ = 637.2309 [M+Na]^+^; found: 637.2322. ^1^H NMR (CDCl_3_, 500.13 MHz, 25 °C): δ = 8.25 (s, 1H, NH-C_4_ cyclo), 7.92 (m, 2H, H_o_, *C_6_H_5_*-CONHC_4_ cyclo), 7.72 (s, 1H, N*H*-C_4_=C_1’_ (vinyl)), 7.66 (m, 2H, H_o_, *C_6_H_5_*-CONHC_q_ (vinyl)), 7.56–7.33 (m, 6H, 2H_p_+ 4H_m_, CO*C_6_H_5_*), 7.23–7.05 (m, 6H, 2H_p_+ 4H_m_, C_6_H_5_), 6.99 (m, 2H, H_o_, =C(H)-*C_6_H_5_*), 6.92 (m, 2H, H_o_, C_3′-_*C_6_H_5_*), 6.50 (d, 1H, =C_1’_H, ^3^J_HH_ = 10.8 Hz), 6.22 (dd, 1H, C_6_H_5_-*CH=CH*, ^3^J_HH_ = 10.2 Hz, ^4^J_HH_ = 2.7 Hz), 6.10 (dd, 1H, C_6_H_5_-*CH=CH*, ^3^J_HH_ = 10.2 Hz, ^4^J_HH_ = 2.1 Hz), 4.68 (t, 1H, H-C_2’_ cyclo, ^3^J_HH4_ = ^3^J_HH3_ = 12 Hz), 4.01 (s, 3H, C_1_-C(O)O*CH_3_*), 3.76–3.70 (m, 4H, =C-C(O)O*CH_3_* + H-C_1’_ cyclo), 3.30 (dd, 1H, H-C_3’_ cyclo, ^3^J_HH_ = 10.20 Hz, ^3^J_HH_ = 12 Hz). ^13^C{^1^H} NMR (CDCl_3_, 125.75 MHz, 25 °C): δ = 171.51 (*C*(*O*)OCH_3_), 166.19 (*CO*NH), 165.33 (2 C(O), *CO*NH + *C(O)*OCH_3_), 142.96 (C_q_, C_6_H_5_), 141.25 (C_q_, C_6_H_5_), 134.32 (C_q_, C_6_H_5_), 134.05 (C_q_, C_6_H_5_), 133.52 (CH, C_6_H_5_), 131.86 (CH, C_6_H_5_), 131.52 (-CH=*CH*-), 128.59 (CH, C_6_H_5_), 128.51 (CH, C_6_H_5_), 128.38 (2CH, C_o_, C_6_H_5_), 128.25 (*C_q_*=CH-), 128.13 (CH, C_6_H_5_), 128.08 (CH, C_6_H_5_), 127.86 (C_q_=*CH*), 127.18 (C_o_, C_6_H_5_), 126.91 (C_o_, C_6_H_5_), 126.71 (-*CH*=CH-), 126.60 (CH, C_6_H_5_), 126.43 (CH, C_6_H_5_), 64.20 (C_4_), 53.30 (CH_3_, C(O)*OCH_3_*), 52.81 (CH_3_, C(O)*OCH_3_*), 51.96 (CH cyclo, C_3’_), 50.33 (CH cyclo, C_1’_), 44.23 (CH cyclo, C_2’_).

#### 3.7.2. Synthesis of Methyl-1-benzamido-2-((E)-2-benzamido-2-methoxycarbonylprop-1-en-1-yl)-3-(4-chlorophenyl)-4-((E)-chlorostyryl)cyclobutane-1-carboxylate **3b**

Compound **3b** was obtained following the experimental procedure described for **3a**. Therefore, oxazolone **1b** (0.2991 mg, 0.968 mmol) and [Ru(bpy)_3_](BF_4_)_2_ (0.036 mg, 0.048 mmol) (5% mol ratio) were irradiated for 18 h in CH_2_Cl_2_ (5 mL) and then reacted with NaOMe in refluxing MeOH (5 mL) for 45 min to give **3b** as a white solid after chromatographic purification and crystallization in CH_2_Cl_2_/*n*-pentane. Obtained: 230.0 mg (78% yield). Mp: 151–152 °C. HRMS (ESI^+^) [*m*/*z*]: calculated for [C_38_H_32_Cl_2_N_2_NaO_6_]^+^ = 705.1535 [M+Na]^+^; found: 705.1524. ^1^H NMR (CDCl_3_, 500.13 MHz, 25 °C): δ = 8.09 (s, 1H, NH, NH-C_4_ cyclo), 7.86 (m, 2H, H_o_, *C_6_H_5_*-CONHC_4_ cyclo), 7.81 (s, 1H, NH, NH-C_4_=C_1’_ (vinyl)), 7.67 (m, 2H, H_o_, *C_6_H_5_*-CONHC_4_), 7.52–7.38 (m, 6H, 2H_p_+ 4H_m_, CO*C_6_H_5_*), 7.16 (dm, 2H, H_m_, C_6_H_4_Cl, ^3^J_HH_ = 8.4 Hz), 7.09 (dm, 2H, H_m_, C_6_H_4_Cl, ^3^J_HH_ = 8.1 Hz), 6.90 (dm, 2H, H_o_, C_6_H_4_Cl, ^3^J_HH_ = 8.4 Hz), 6.85 (dm, 2H, H_o_, C_6_H_4_Cl, ^3^J_HH_ = 8.1 Hz), 6.64 (d, 1H, =C_1’_H oxazolone, ^3^J_HH_= 11.9 Hz), 6.13 (dd, 1H, *CH=CH*-C_4_, ^3^J_HH_= 10.1 Hz, ^4^J_HH_= 2.7 Hz), 6.10 (dd, 1H, -*CH=CH*-C_4_, ^3^J_HH_= 10.1 Hz, ^4^J_HH_= 2.1 Hz), 4.75 (t, 1H, H-C_2’_ cyclo, ^3^J_HH_ = ^3^J_HH_ = 11.9 Hz), 4.00 (s, 3H, C_1_-C(O)O*CH_3_*), 3.69–3.64 (m, 4H, =C-C(O)O*CH_3_* + H-C_1’_ cyclo), 3.27 (dd, 1H, H-C_3’_ cyclo, ^3^J_HH_ =10.4 Hz, ^3^J_HH_ = 11.9 Hz). ^13^C{^1^H} NMR (CDCl_3_, 125.7 MHz, 25 °C): δ = 171.53 (C(O), *C*(O)OCH_3_), 166.23 (C(O), *C*ONH), 165.57 (C(O), *C*ONH), 165.16 (*C*(O)OCH_3_), 141.42 (C_i_, C_6_H_4_Cl), 139.83 (C_i_, C_6_H_4_Cl), 134.38, 134.30 (C_i_, C_6_H_5_ + C_i_, C_6_H_5_), 133.17 (-*C*H=CH-), 132.57, 132.54 (C_p_, C_6_H_4_Cl + C_p_, C_6_H_4_Cl), 132.13 (C_p_, C_6_H_5_), 131.83 (C_p_, C_6_H_5_), 129.84 (C_o_, C_6_H_4_Cl), 129.79 (C_o_, C_6_H_4_Cl), 128.83, 128.74 (C_m_, C_6_H_5_ + C_m_, C_6_H_5_), 128.63 (C_m_, C_6_H_4_Cl), 128.59 (C_m_, C_6_H_4_Cl), 128.40 (*C*_q_=CH), 127.27 (2C, C_o_, C_6_H_5_ + C_q_=*C*H), 127.17 (-CH=*C*H-), 127.06 (C_o_, C_6_H_5_), 64.39 (C_4_, cyclo), 53.69 (CH_3_,C(O)O*C*H_3_), 53.09 (CH_3_, C(O)O*C*H_3_), 51.71 (CH cyclo, C_3’_), 49.78 (CH cyclo, C_1’_), 43.88 (CH cyclo, C_2’_).

## 4. Conclusions

The Ru-sensitized irradiation of 4-(3-aryl-allyliden)-5(4*H*)-oxazolones with blue light (456 nm) takes place with the formation of a series of styryl-cyclobutanes by [2+2]-photocycloaddition of two oxazolones. The coupling is produced using the exocyclic C=C bond of one oxazolone and the styryl C=C bond of the other one and occurs with a high degree of selectivity, because one isomer is mostly formed. This coupling is very different from that observed in 4-aryliden-5(4*H*)-oxazolones, which involves only the exocyclic C=C bond. The reactive species was characterized as a triplet excited state T_1_ by transient absorption spectroscopy. In addition, DFT methods showed that this T_1_ state is planar and that the spin distribution is concentrated in the α and δ carbons of the diene system, instead of in the α and β carbons as observed in aryliden-oxazolones, providing a reasonable explanation for the different reactivity observed.

## Figures and Tables

**Figure 1 ijms-24-07583-f001:**
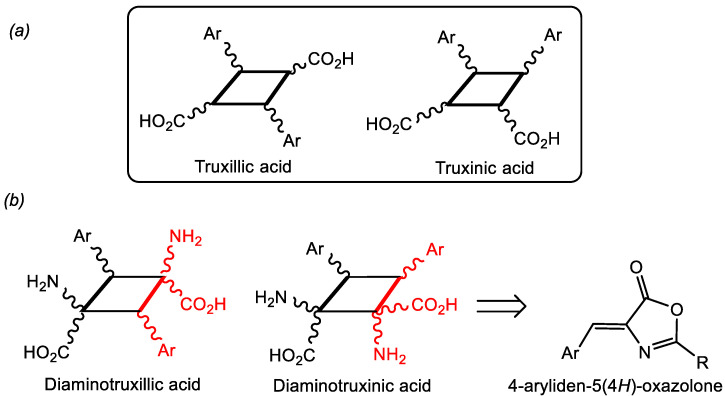
Schematic representation of (**a**) truxillic and truxinic acids; (**b**) unnatural 1,3-diaminotruxillic and 1,2-diaminotruxinic amino acids, showing their simplest retrosynthetic route.

**Figure 2 ijms-24-07583-f002:**
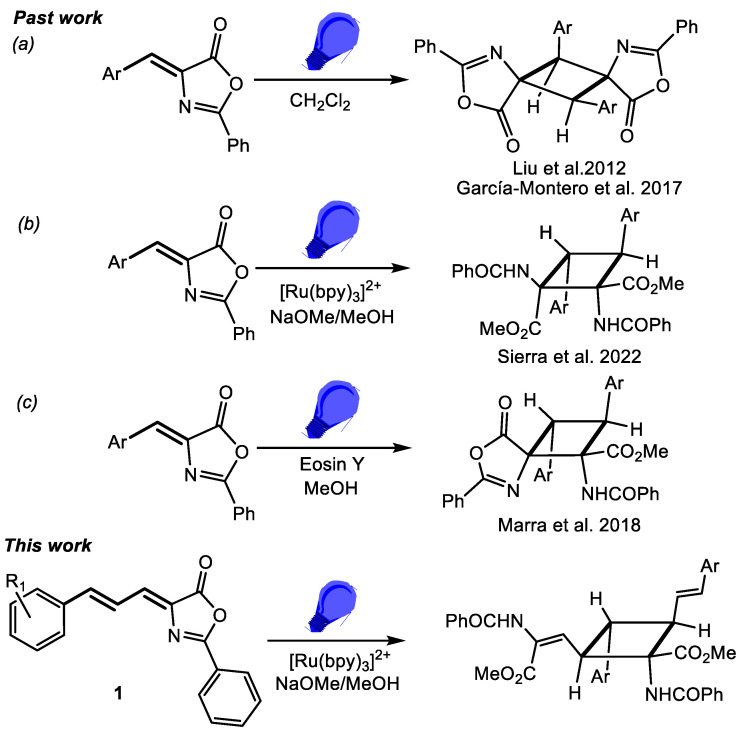
Past work reported on the synthesis of truxillic and truxinic derivatives from irradiation of aryliden-oxazolones: (**a**) direct irradiation of oxazolones giving 1,3-coupling and the epsilon isomer; (**b**) irradiation in presence of a Ru-photosensitizer giving 1,2-coupling and the mu-isomer; (**c**) irradiation in presence of an organic photosensitizer giving 1,2-coupling and the zeta-isomer [26,30,31,32]. Comparison with the present work from allyliden-oxazolones.

**Figure 3 ijms-24-07583-f003:**
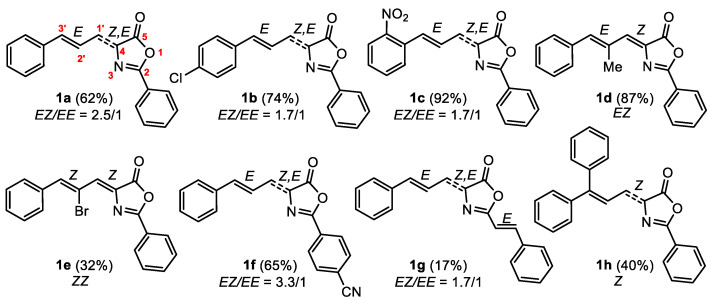
2-aryl-4-(*E*-3′-aryl-allylidene)-5(4*H*)-oxazolones **1a**–**1h** prepared for this work.

**Figure 4 ijms-24-07583-f004:**
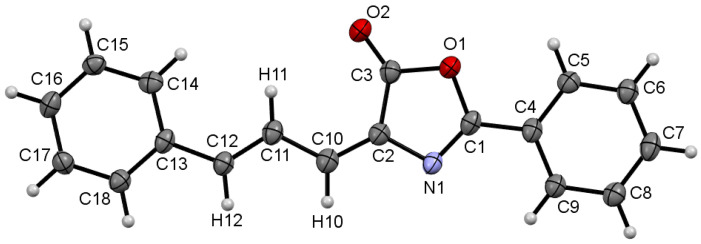
ORTEP drawing of oxazolone **1a** showing the minor isomer with *EE*-configuration. Thermal ellipsoids are drawn with 50% probability.

**Figure 5 ijms-24-07583-f005:**
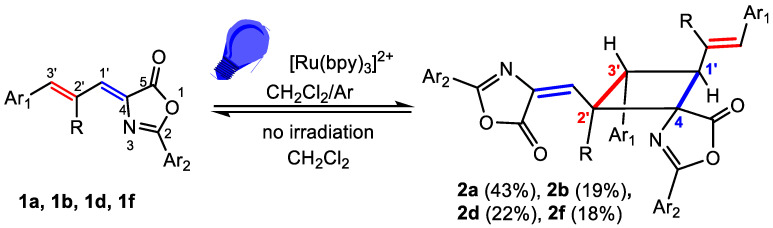
Selective formation of styryl-cyclobutanes **2** from allylidene-5(4*H*)-oxazolones **1**.

**Figure 6 ijms-24-07583-f006:**
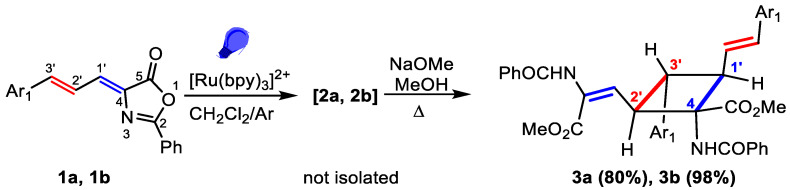
Selective formation of bis-amino acids **3** from allylidene-5(4*H*)-oxazolones **1** in a one-pot, two-steps synthesis.

**Figure 7 ijms-24-07583-f007:**
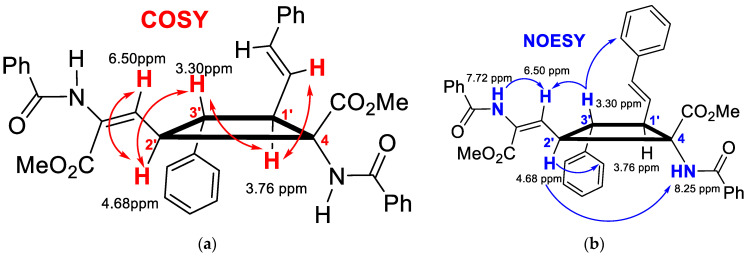
(**a**) Key correlations observed in the ^1^H-COSY spectrum of **3a** (Appendix A; (**b**) Key correlations observed in the ^1^H NOESY spectrum of **3a** (Appendix A).

**Figure 8 ijms-24-07583-f008:**
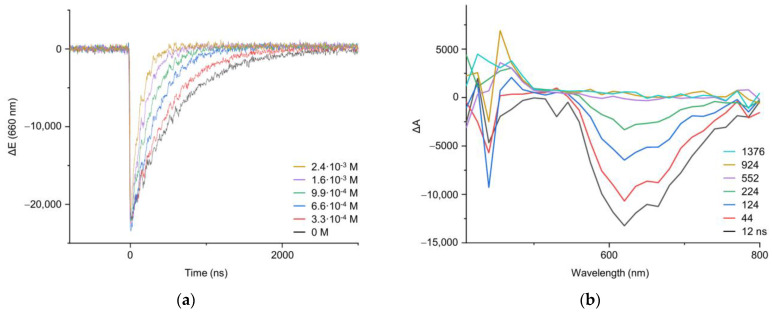
(**a**) Decay traces recorded at 660 nm for [Ru(bpy)_3_]^2+^ (in deoxygenated CH_2_Cl_2_) after addition of different amounts of **1a**, obtained after LFP excitation at 532 nm; (**b**) Transient absorption spectrum of a deoxygenated CH_2_Cl_2_ solution of [Ru(bpy)_3_]^2+^ in the presence of **1a** (2.4 · 10^−3^ M) registered at different times after laser pulse (λ_exc_ = 532 nm).

**Figure 9 ijms-24-07583-f009:**
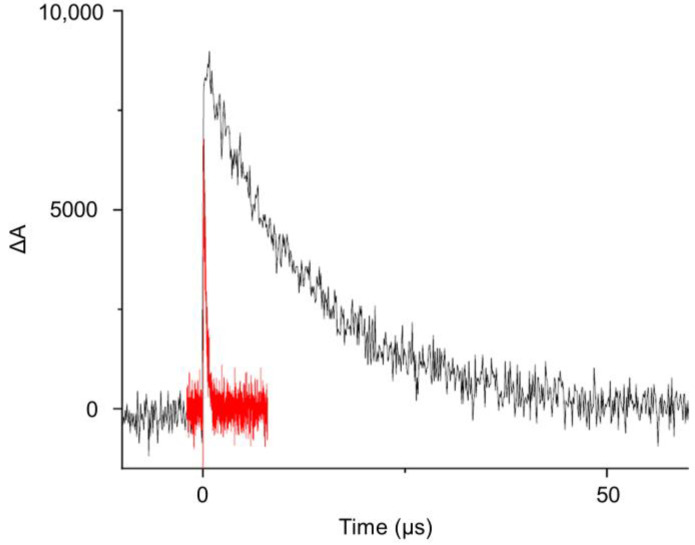
Transient absorption traces recorded at 470 nm upon LFP excitation (532 nm) of [Ru(bpy)_3_]^2+^ in the presence of **1a** (4.7 · 10^−3^ M) in deoxygenated (black) and oxygenated CH_2_Cl_2_ (red).

**Figure 10 ijms-24-07583-f010:**
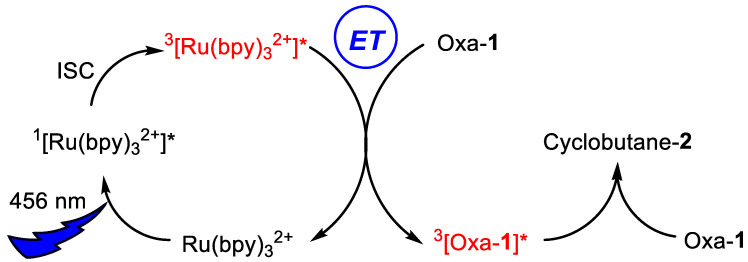
Proposed mechanism for the formation of the reactive species of the oxazolone ^3^[oxa-**1**]* by photosensitization of the oxazolone in the ground state oxa-**1** from ^3^[Ru(bpy)_3_^2+^]* by energy transfer. Further reaction of ^3^[oxa-**1**]* with oxa-**1** affords the cyclobutanes **2**.

## Data Availability

Data is contained within the article or Appendix A.

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
