# Peer review of "Synthesis of Bis(amino acids) Containing the Styryl-cyclobutane Core by Photosensitized [2+2]-Cross-cycloaddition of Allylidene-5(4H)-oxazolones"

_ijms, 2023, doi:10.3390/ijms24087583_

Round 1
Author Response
Reviewer: 1
Comments: major revision
This manuscript report on the photodimerization by [2+2]-cross-cycloaddition of allylidene-5(4H)-oxazolones affording bis(amino acids) containing the styryl-cyclobutane core. It represents an implementation on previous photochemical dimerization studies of arylidenoxazolones but a new class of non-natural bis-amino acids containing the decorated cyclobutane scaffold were obtained. Since the topic is the synthesis of new molecules, without any biological connection, it not seems suitable for Int. J. Mol Sci. but it can be considered for publication on Molecules after major revisions as below indicated.
I suggest to point out in the introduction on the general importance of the synthesis of new non-natural amino acids because your new compounds are neither truxillic nor truxinic derivatives.
- R) A short paragraph reporting the interest of the development of new strategies for the synthesis of new unnatural amino acids, with the corresponding references, has been included. All references have been renumbered accordingly.
As main address, I suggest to insert the catalytic cycle for this photodimerization (see below the comments.
- R) A proposal for catalytic cycle has been introduced in section 2.3 as requested. This catalytic cycle is shown in Figure 10, so figures ahead this one have been renumbered
Pg 2 Figure 1: delete the acrylate that is not an intermediate in the synthesis of diamminotruxinic acid
Pg 2 Figure 2 (and Figure 6)
- Avoid inserting parenthesis in the substituent (es N(H)C(O)Ph change with NHCOPh)
- This work: insert number 1 under the oxazolone
Pg 2 line 62 Wang and coworkers irradiated a couple of arylidene-oxazolones
- R) All suggested changes in this page have been perfomed
Pg 3 line 70 Change with ([Ru(bpy)3](BF4)2, bpy = 2,2'-bipyridine, versus Eosin Y, Figures 2b and 2c)
Pg 3 line 74 “This fact is quite noteworthy because the reactions occur in solution and in absence of chiral auxiliaries”. Your result is excellent since it is possible to control the regio- and diastereoselectivity. On the other hand, it is not the presence of a chiral auxiliary that necessarily drives the regio- diastereo control of this reaction. So please, reformulate the sentence.
Pg 3 line 86 Change this sentence with: “the possibility to control the regio- and stereoselectivity, as the number of C=C double bonds increases, is highly challenging
- R) The suggested changes and reformulations have been perfomed in the text
Pg 3 line 96: I suggest to avoid E and Z because you have an example with Z,Z geometry. Please correct with Synthesis of 2-aryl-4-(E-3’-aryl-allylidene)-5(4H)-oxazolones (correct also the name in line 87)
- R) The suggested change has been done
Pg 3, Figure 3. To better understand the NMR description, I suggest to indicate the numbering in one formula of Figure 3, as shown in Figure 5.
- R) Numbering has been introduced in Figure 3
Pg 3 line 106: I suggest to change this sentence in “All oxazolones were obtained in moderate to good yields, except oxazolone 1g which is highly soluble in ethanol (see experimental for details)”
- R) The requested change has been done
Pg 4 lines 111-121:
- not clear the meaning of “inversion” of the signal, do you intend after irradiation? R: there are two ways to promote NOE, saturation (NOEdiff) and inversion (NOESY), in this case we have measured 1D-NOESY, so an initial inversion of the signal is produced). For the sake of clarity, instead of inversion, we have changed the word for the more general "perturbation"
- change with: to the Horto (or Ho) of the 3'-Ph ring (see also experimental)
- R) Change done
- I suppose there are several mistakes in indicating numbers in the double bond: the configuration of the C1'=C2' double bond is E in 1d: Correct with C3'=C2'; The configuration of the exocyclic C3'=C double bond; Correct with C1’=C4
- R) Reviewer is totally right, all assignations were wrong, thank you very much for the hard work
- Correct with: with 3JH3'H2' values identical in the two isomers (15.7 Hz), suggesting a Econfiguration of the C3'=C2' double bond in both isomers
- R) Change done
- I suggest to change the following sentence: “As expected, the large 3JH1'H2' value (11.6 Hz) suggests the formation of the s-trans rotamer of the diene system.
- R) Change done
Figure 4. Add minor isomer in the legend
- R) Change done
Pg 4: My suggestion is to move the X-Ray explanation in the Supporting
- R) This is probably the only item in which we don't agree with the reviewer, since the X-ray structure confirms the NMR discussion, provides additional information about electronic delocalization and planarity in the reactive fragments of the oxazolone, and helps for the building of a model in the ground state. Moreover, the short description given in the text extends only 9-10 lines, so it is just a short section in the paper. Therefore, this short paragraph provides valuable information and we think that it merits to be in the main text of the article.
- There is a mismatch in defining the C1’=C4 geometry (sometimes Z/E others E/Z). For example, in figure 3 you attributed the sequence EZ/EE, that means E for the aryl substituted double bond and Z to the exocyclic. Same sequence when you described 1d and 1e, but not when you described the major and minor isomers (Pg 4 lines 128-134) Please uniform. Check also carefully the experimental part because there are a lot of these mismatches
- R) Totally agree with the reviewer, and thank you for the correction. We have adopted the nomenclature shown in Figure 3, in such a way that the first descriptor represents the C3'=C2' alkene, while the second one represents the C1'=C4. We have checked the whole manuscript several times, and we think that now all is consistent with this system.
Pg 5 Figure 5: Please add the equilibrium arrows in the scheme indicating the transformation of
compounds 2 into 1 when in solution.
- R) The requested change has been done
Furthermore, the numbering of cyclobutane ring does not correspond to the real number, as reported in the experimental section. Please correct (also in the text). Furthermore, my suggestion is to indicate the numbers in the double bond as previously reported in the oxazolone (1’, 2’ 3’). This must also be done in the experimental section (NMR description): it is quite difficult to assign the correspondence between the chemical shift and the corresponding proton. In some cases, there is more than one proton with the same number (for example see NMR of 1c).
- R) We have followed the suggestion of the reviewer, and we have modified accordingly figures 5 and 6, and the new figure 7, showing the new numbering scheme. This new numbering scheme has also been applied to the experimental section. We have checked all cases and we think that now all is consistent.
Pg 5. Lines 190-192 Change the sentence as indicated: The simplest strategy to eliminate this steric constrain and to obtain stable cyclobutane derivatives is the transformation of oxazolone ring in compound 2 into the corresponding ester 3 ( Figure 6).
- R) The requested change has been done
Pg 6 line 202: following a procedure identical to that shown in Figure 5 change with as reported above
- R) The requested change has been done
Pg 6 NMR sentence regarding compounds 2 “but the absence of significative NOE cross-peaks precludes the …” If you consider the J values you can define the trans disposition of C-2, C-3 and C-4 substituents. On the other hand, it is quite strange to me you can not see spatial proximities in the Noesy experiment (not reported in SI)!
- R) The NOESY of 2d was provided in the original supplementary material (Figure S61), there it was clear that the information provided is limited, mostly when compared with the information provided by the NOESY of 3a, which has been fully analyzed.
Concerning compound 3a, to help the reader to check your statements in the text, I suggest to indicate the proton corresponding to the specific signal in the NOESY plot in the supporting information and call back the figure number in the text (Figure FS????). Pg 6 line 224-230: focusing on the cylobutane ring, change the numbering according to the above proposal. There are three sentences (i), ii), and iii)) that are not documented by experimental data. Please define carefully which are the data (es J, NOEs) supporting your conclusion on the sterochemistries.
- R) We have added in the text the full reasoning that we performed to determine the structures of these compounds. We have included a new Figure (7) containing the most important COSY and NOESY correlations , with the assignation of the protons involved and with reference to the corresponding figures in the supplementary material. We hope that all these incorporations help to understand the non-trivial structural characterization of these cyclobutanes.
Paragraph 2.3. I propose to insert a catalytic cycle considering you hypothesize a mechanism via energy transfer and not electron transfer (therefore redox process) according to the Stern-Volmer analyses, mechanism corroborated by DFT calculations.
- R) A proposal for catalytic cycle has been introduced in section 2.3 as requested. This catalytic cycle is shown in Figure 9, so figures ahead this one have been renumbered.
In general, when you refer to a figure/s indicated in the Supporting, insert the number of the figure/s
(FS1 etc). Please insert also in the supporting FS1 etc…(you can anticipate this part in SI before NMR spectra)
- R) The requested format has been included
Pg 6 line 239: “instead of with another identical C=C bond” change with “instead of with an other
exocyclic C=C bond”
- R) Done
Pg 6 line 241: alkenes
- R) Corrected
Pg 7. Please indicate the axes legend in English (Figure 7 and 8 and also the figures in the SI) as well as insert a measure units better defined. Furthermore, in the legend of 7b) there are two blu lines (124 ns and 1376 ns), one of them must be "dark blue", (see SI for 1b e 1d)
- R) Legends have been changed to english, both in the supplementary material and in the main text, as requested. In the case of Figure 8b, for the sake of clarity, we have included the values of the times in the figure, instead of in the legend of the figure.In this way there are no ambiguity between the values and the blue colors.
Pg 9 Figure 9: B T1 rotation: In the first plot the scale numbers were omitted. Can you show also the part over 180°?
- R) We would like to thank the reviewer for their observations. The scale numbers were deliberately excluded for the sake of simplicity since the labels are already included in the bottom plot, which uses the same scale as the top plot. Additionally, to ensure clarity, we included a sentence explaining that any values above 180° are converted into their equivalent values between 0 and 180°: “Since rotations are symmetrical in both directions, any values above 180° were converted to their equivalent degrees between 0 and 180° for clarity (i.e., 190° would be equivalent to 170°).”
Experimental
- a) As suggested above, check all EZ/Z/E and uniform
Ra) As explained above, we have checked this nomenclature throughout the text
- b) When you indicate the ration of isomers of 1, you use a different presentation for the different
compounds. (es: Ratio 1:0.4 (Z,E/E,E)); Ratio 1:0.6 (Isomer (E,Z):Isomer (E,E)) Please uniform
Rb) This has been changed in all compounds 1
- c) Is there any reason because the stoichiometries of the reagents change for the synthesis of
compounds 1? It seems very strange to me.
Rc) The molar ratio of aldehyde / hippuric / acetate has to be 1 / 1 / 1, and this is accomplished in all cases except in 1f, where an error has been detected and corrected. Thank you for the observation.
- d) Insert a general procedure for the synthesis of both 1 and 2 and for the specific substrate its
specific data avoiding repetitions.
Rd) We have minimized the information for each substrate, but we think that the amounts of reactants are important data to be given, together with HRMS and NMR data.
- e) Except for 1d,f, compounds 1 were obtained as mixtures of isomers, not separated. On the other
hand, the NMR description for the minor isomer is reported only for 1a. I suggest to insert the main
chemical shifts for the minor isomers in the other cases.
Re) The NMR description of the minor isomer EE was given in the cases of 1a and 1c, because they were the only cases where all signals could be identified unambiguously. This can not be done in the other cases, due to extensive overlapping of most of the peaks, including the relevant H1', H2' and H3' protons. In order to uniformize the information given for all compounds, we have removed the 1H NMR data of the minor isomer of 1a and 1c.
- f) In general, there are many multiplets in NMR description indicated as a single chemical shift.
Indicate the interval.
Rf) An interval of chemical shifts is given when several peaks overlap and it is not possible to separate the individual contributions of each peak. However, when several peaks overlap but it is possible to distinguish the individual contributions of each peak, then separated chemical shifts have been given with the shape of each signal. We have revised all experimental section, peak by peak, using this methodology, and we have corrected several errors.
- g) compound 1e: This is the Z,Z-isomer
Rg) Changed
- h) Oxazolone 1f: as for the other examples, insert mg and not ml of cinnamaldehyde. Molecular formula of 1f: correct with H12
Rh) Mg of cinnamaldehyde have been changed in 1f and 1g. With respect to the molecular formula of 1f, what was observed in the MS was the protonated ion [M+H], therefore H13 is correct (in other cases the M+Na ion was observed): for the sake of clarity, we have added in the text that C19H13N2O2 corresponded to the [M+H] positive ion
- i) Oxazolone 1h: in the main text you indicated the formation of a mixture of isomer but not in the experimental: insert the ratio
Ri) No mixtures were detectd in the case of 1h, this has been corrected in the text
- j) Compounds 2: Change number for the NMR protons (H1, H2’,H3’) as suggested above.
Rj) The requested change has been done
- k) The number of protons reported in NMR description for 2a is 24 and not 26
Rk) This has been corrected
- l) 2f: NMR signal d 3.79-3.70: indicate the number of protons
Rl) This has been corrected
- m) 3a: 3-methoxy-3-oxo change with 2-methoxycarbonyl
Rm) This has been corrected
- n) Are compounds 3a and 3b pure isomers? If so, please indicate the melting point. Change the number for the olefinic protons according to the suggestions above.
Rn) Melting points of 3a and 3b have been measured and included in the experimental section
Supplementary material
- a) Standardize legends for the absorption spectra of oxazolones in the Supporting Information (image title, axes, etc.)
Ra) All those aspects have been revised
- b) Insert Figure number (FS1 etc.). Check EZ/ZE; legends blu/darkBlu; deaerated/areate/deoxigenated; Spanish/English
Rb) All those aspects have been revised

Reviewer 2 Report
In this work the authors reported the synthesis of bis(amino acids) containing the styryl-cyclobutane core by photosensitized [2+2]-cross-cycloaddition of allylidene-5(4H)-oxazolones. The authors verified that the different reactivity of photocatalyst were caused by additional C=C bond conjugated with exocyclic C=C bond with the aid of transient absorption spectroscopy and DFT modeling. Considering recent great interest in discovering and synthesizing novel bis(amino acids), I think the topic of this work is timely and the findings presented are very interesting. Overall the paper is well written, and I recommend the publication of this work in IJMS. I have a few minor comments that can hopefully help the authors to improve their paper.
The following points need to be addressed:
1. In introduction, the authors give a mini review about the synthetic method in Figure 2, I noted that even it is the simplest method, only four works have been reported, why? What’s the drawback about this method? And what is the frequently used? Could the author give some discussion about it?
2. In Figure 7(b), there are too many details especially in the beginning, could the author offer a zoomed-in view and give more explanation about the differences among different time scales?
3. In the discussion about 1a and 4’s molecular dynamics, the author referred to the low barriers about 2.6 and 2.5 kcal·mol-1. Altogether, this is a bit concerning. What are the typical barriers interconvert from E to Z structures in literature? Are those acceptable?
4. The General procedure for the synthesis of the oxazolones are too long, to keep the most important points and move the rest to SI may be better.
The present manuscript contains a few spelling and grammar mistakes. Please check it carefully.

Author Response
Reviewer 2
Comments: publication
In this work the authors reported the synthesis of bis(amino acids) containing the styryl-cyclobutane core by photosensitized [2+2]-cross-cycloaddition of allylidene-5(4H)-oxazolones. The authors verified that the different reactivity of photocatalyst were caused by additional C=C bond conjugated with exocyclic C=C bond with the aid of transient absorption spectroscopy and DFT modeling. Considering recent great interest in discovering and synthesizing novel bis(amino acids), I think the topic of this work is timely and the findings presented are very interesting. Overall the paper is well written, and I recommend the publication of this work in IJMS. I have a few minor comments that can hopefully help the authors to improve their paper. The following points need to be addressed:
- In introduction, the authors give a mini review about the synthetic method in Figure 2, I noted that even it is the simplest method, only four works have been reported, why? What’s the drawback about this method? And what is the frequently used? Could the author give some discussion about it?
R1) Even if it sounds strange, there are only four main contributions about synthesis of cyclobutanes by direct irradiation of oxazolones and subsequent 2+2 photocycloaddition. We have not considered our previous contributions using Pd templates (Chem Commun 2009, 4681; Chem. Eur. J. 2016, 22, 144 ; Eur. J. Inorg. Chem. 2019, 3481; Beilstein J. Org. Chem. 2020, 16, 1111) because in the present contribution Pd is not involved. We have neither considered one single example published by Sampedro and Runcevski (Angew. Chem. Int. Ed. 2014, 53, 6738) because the reaction takes place in solid state under topochemical conditions, far from the present case. In the same respect, photocycloaddition of thiazolones published by our group (J. Org. Chem. 2021, 86, 12119) was not included because the corresponding bis-amino acids were finally not obtained. Therefore, if we focus in direct irradiation of oxazolones to give cyclobutanes through [2+2], there are only the examples given.
Why the reaction is scarcely represented? In our opinion (this is just an hypothesis) because, in fact, direct irradiation was already studied using UV light resulting in Z-E isomerization (Diego Sampedro and coworkers have studied in detail this reaction). Surprisingly (luckly for us) the irradiation with visible light (blue) followed a different path, the photocycloaddition, and this was unexpected. Under our point of view there are no drawbacks, just the reactions take place slowly, but in general are quite tolerant with the substituents and are very stereoselective.
- In Figure 7(b), there are too many details especially in the beginning, could the author offer a zoomed-in view and give more explanation about the differences among different time scales?
R2) Figure 7b (now 8b) contains two relevant facts, which are the disappearance of the intense phosphorescence of Ru at 620 nm (quenching of triplet) and the simultaneous appearance of a weak absorption at 450 nm due to the generated triplet of the oxazolone. At short times after pulse (black line, 12 ns) the main speces is the triplet of the Ru, while at long times (552 or 924 ns, grey and ochre lines) the triplet of the Ru has almost disappeared due to energy transfer to the oxazolone and subsequent formation of its triplet excited state, which is visible in the emerging absorption at 450 nm. Both facts, and their interpretation, are explained in the text and can be clearly seen in the figure 8b. However, for the sake of clarity, we have added a pair of new sentences explaining the figure. Moroever, this figure allows the visualization of both facts at the same time, so we think that it is really illustrative and the best way to understand the process at the excited states. Zooming in one of the regions gives only partial information and the whole perspective of the process will be missed. Due to all those facts we left figure 8b unaltered, just translating the legend to english
- In the discussion about 1a and 4’s molecular dynamics, the author referred to the low barriers about 2.6 and 2.5 kcal·mol-1. Altogether, this is a bit concerning. What are the typical barriers interconvert from E to Z structures in literature? Are those acceptable?
R3) We have taken into account the reviewer's concern and added a more comprehensive discussion on this topic in the main text. In this new discussion, we have included an experimental reference (DOI: https://pubs.acs.org/doi/full/10.1021/acs.joc.1c03092) that demonstrates how different isomers result in the same triplet structure and undergo rapid isomerization, which is consistent with the calculated low barriers: “The low calculated barriers are consistent with previous photophysical experimental results, which indicate that the same triplet state is rapidly formed when starting from the E and Z forms of compound 4. [32] These rapid isomerization processes may be an important contributing factor to the low yields obtained experimentally since multiple competitive isomers of 2 can be formed.”
- The General procedure for the synthesis of the oxazolones are too long, to keep the most important points and move the rest to SI may be better.
R4) We have changed the experimental section of the oxazolones, keeping the minimum text for all cases (color of the solid, amounts of reagents, HRMS and NMR data) and giving a detailed description only for the first one (this one has also been condensed). We prefer to keep in the manuscript the experimental details because this give a more solid consistence to the article
- The present manuscript contains a few spelling and grammar mistakes. Please check it carefully.
R5) We have made our best in this respect, aiming to correct as much as possible mistakes.
